# Menstrual Cycle Changes Joint Laxity in Females—Differences between Eumenorrhea and Oligomenorrhea

**DOI:** 10.3390/jcm11113222

**Published:** 2022-06-05

**Authors:** Sae Maruyama, Chie Sekine, Mayuu Shagawa, Hirotake Yokota, Ryo Hirabayashi, Ryoya Togashi, Yuki Yamada, Rena Hamano, Atsushi Ito, Daisuke Sato, Mutsuaki Edama

**Affiliations:** 1Institute for Human Movement and Medical Sciences, Niigata University of Health and Welfare, Shimami-cho 1398, Niigata City 950-3198, Japan; hpm20010@nuhw.ac.jp (S.M.); sekine@nuhw.ac.jp (C.S.); hpm21008@nuhw.ac.jp (M.S.); yokota@nuhw.ac.jp (H.Y.); hirabayashi@nuhw.ac.jp (R.H.); hpm21012@nuhw.ac.jp (R.T.); hpm21016@nuhw.ac.jp (Y.Y.); daisuke@nuhw.ac.jp (D.S.); 2Department of Health and Sports, Niigata University of Health and Welfare, Shimami-cho 1398, Kita-ku, Niigata City 950-3198, Japan; hamano@nuhw.ac.jp (R.H.); atsushi-ito@nuhw.ac.jp (A.I.)

**Keywords:** anterior knee laxity, stiffness, general joint laxity, genu recurvatum

## Abstract

The purpose of this study was to investigate the changes in anterior knee laxity (AKL), stiffness, general joint laxity (GJL), and genu recurvatum (GR) during the menstrual cycle in female non-athletes and female athletes with normal and irregular menstrual cycles. Participants were 19 female non-athletes (eumenorrhea, *n* = 11; oligomenorrhea, *n* = 8) and 15 female athletes (eumenorrhea, *n* = 8; oligomenorrhea, *n* = 7). AKL was measured as the amount of anterior tibial displacement at 67 N–133 N. Stiffness was calculated as change in (Δ)force/Δ anterior displacement. The Beighton method was used to evaluate the GJL. The GR was measured as the maximum angle of passive knee joint extension. AKL, stiffness, GJL, and GR were measured twice in four phases during the menstrual cycle. Stiffness was significantly higher in oligomenorrhea groups than in eumenorrhea groups, although no significant differences between menstrual cycle phases were evident in female non-athletes. GR was significantly higher in the late follicular, ovulation, and luteal phases than in the early follicular phase, although no significant differences between groups were seen in female athletes. Estradiol may affect the stiffness of the periarticular muscles in the knee, suggesting that GR in female athletes may change during the menstrual cycle.

## 1. Introduction

Injury to the anterior cruciate ligament (ACL) is a sports injury that occurs more frequently in women than in men [1]. Sex differences in the incidence of ACL injury are attributed in part to the influence of female hormones, the concentrations of which fluctuate during the menstrual cycle [2]. The menstrual cycle is regulated by the female hormones estradiol and progesterone, and is divided into follicular, ovulation, and luteal phases. ACL injury is reportedly more likely to occur during the follicular [3] and ovulation phases [4]. Clarifying the relationship between risk factors for ACL injury and female hormones is therefore an important step in minimizing the risk of such injury.

One potential risk factor for ACL injury in women is joint laxity. Joint laxity has been examined in terms of anterior knee laxity (AKL), general joint laxity (GJL), and genu recurvatum (GR), each of which have been reported to show associations with risk of ACL injury [5,6,7]. A previous study found that AKL [8,9], GJL [10], and GR [11] were all higher in women than in men. In addition, AKL, GJL, and GR may be altered by changes in the concentrations of female hormones during the menstrual cycle [9,12,13]. Previous studies therefore suggest that changes in joint laxity due to changes in female hormone concentrations may be related to differences in the timing of ACL injury during the menstrual cycle.

Some reports have suggested that AKL, which indicates the laxity of the ACL, is altered by changes in hormone levels during the menstrual cycle [9,12,14], while others have reported the AKL shows no significant change [8,15,16,17,18]. Likewise, studies have reported both that GJL changes [12,18] and that GJL remains unchanged during the menstrual cycle [17]. Such contradictory findings may be due to the fact that the timings of measurements and subjects differed between studies.

In addition, most previous studies investigating changes in joint laxity during the menstrual cycle have been conducted among women with normal menstrual cycles. Lee et al. examined the effect of oral contraceptive (OC) use on AKL and found that AKL was significantly higher in the non-OC group than in the OC group, and that AKL was increased during the ovulation and luteal phases compared with the early follicular phase [19]. Decreases in levels of female sex hormones due to OC use may thus affect AKL. In addition, it has been reported that 20–30% of Japanese female athletes reported experience menstrual irregularities [20,21] such as oligomenorrhea [22]. To investigate the effects of female hormones on joint laxity in greater detail, women with menstrual irregularities would need to be included in the study. However, to the best of our knowledge, no reports have examined changes in joint laxity during the menstrual cycle among women with menstrual irregularities. Clarification of these issues would contribute to the development of better methods for training and prevention of ACL injuries in accordance with menstrual cycle conditions.

The purpose of this study was to determine changes and differences in joint laxity (AKL, GJL, and GR) during the menstrual cycle among female athletes and non-athletes with normal and irregular menstrual cycles. We hypothesized that AKL, GJL, and GR would all change during the menstrual cycle in female athletes and non-athletes with normal menstrual cycles, and would not change during the menstrual cycle in female athletes and non-athletes with menstrual irregularities.

## 2. Materials and Methods

### 2.1. Participants

Seventy-one female non-athletes and 27 female athletes affiliated with the university were recruited between July 2020 and July 2021 and administered a questionnaire on inclusion criteria. Inclusion criteria were as follows: (1) no history of injury or surgery involving the osteochondral surfaces, ligaments, tendons, capsule, or menisci of either knee joint; and (2) no use of OCs or other hormonal medications within the 6 months preceding the first day of measurement [23]. Female non-athletes were defined as female who were physically active less than 3 times per week [17]. Non-athletes 19 people (Figure 1) and athlete 15 people (volleyball 10 people, basketball 5 people) (Figure 2) met the inclusion criteria and agreed to participate in the study. This study was approved by the ethics committee at Niigata University of Health and Welfare (approval no. 18467). This study complied with the tenets of the Declaration of Helsinki and was conducted only after obtaining written consent from potential study participants who had been fully informed (in both oral and written form) of the nature of the experiments.

### 2.2. Classification of the Menstrual Cycle

Participants were classified into two groups: a eumenorrhea group and an oligomenorrhea group. The eumenorrhea group was defined to include females with menstrual cycles of 25–38 days [22] for the two cycles before and during the experiment, and at least 10 menstrual cycles in the past 12 months [24]. The oligomenorrhea group was defined to include females with menstrual cycles of either less than 24 days or more than 39 days [22] in the menstrual cycle before or during the experiment, or who had nine or fewer menstrual cycles in the past 12 months [24]. The 19 female non-athletes included 11 individuals in the eumenorrhea group (mean age, 21.0 ± 0.7 years; height, 159.3 ± 4.8 cm; weight, 50.1 ± 7.4 kg; cycle length, 31.3 ± 2.1 days) and 8 individuals in the oligomenorrhea group (mean age, 21.3 ± 0.4 years; height, 158.3 ± 4.2 cm; weight, 54.7 ± 9.2 kg; cycle length, 35.4 ± 7.5 days). The 15 female athletes included 8 individuals in the eumenorrhea group (mean age, 18.8 ± 0.7 years; height, 167.2 ± 6.6 cm; weight, 65.8 ± 10.0 kg; cycle length, 31.0 ± 2.5 days) and 7 individuals in the oligomenorrhea group (mean age, 19.3 ± 0.7 years; height, 167.8 ± 5.7 cm; weight, 60.1 ± 3.6 kg; cycle length, 45.0 ± 12.0 days).

### 2.3. Menstrual Cycle Recording

Participants were asked to measure their basal body temperature (BBT) with a basal thermometer (Citizen Electronic Thermometer CTEB503L; Citizen Systems Co., Ltd., Tokyo, Japan) every morning for 1–3 months preceding the first day of measurement. To estimate the day of ovulation, participants were given an ovulation test kit (Doctor’s Choice One Step Ovulation Test Clear; Beauty and Health Research, Torrance, CA, USA) and asked to use it from the day after the end of menstruation. Participants were asked to record daily BBT, ovulation test kit results, and start and end dates of menstruation in the ONE TAP SPORTS, an athlete’s condition management system (Euphoria Co., Ltd., Tokyo, Japan).

### 2.4. Timing of Measurements

AKL, stiffness, GJL, and GR were measured a total of eight times each on two consecutive days during each of the four phases of the menstrual cycle (early follicular, late follicular, ovulation, and luteal phases), and the average of values from those two days was used as the measurement value for each phase. Salivary hormone levels were measured only on the first day of each phase. Measurements were taken on two days between 2–4 days after the onset of menstruation in the early follicular phase, on two days between 2–4 days after the end of menstruation in the late follicular phase, on two days between 2–4 days after a positive result from the ovulation test kit day in the ovulation phase, and on two days more than 2 days after the transition to the high-temperature phase of BBT or 1 week after the first measurement in the ovulation phase in the luteal phase. In the luteal phase, if BBT during the 3 days after the estimated ovulation day was at least 0.2 °C higher than the average BBT during the first 6 days of menstruation, the BBT was considered biphasic, indicating a transition from the low- to the high-temperatures phase [25]. All measurements were performed between 07:00 and 12:00 to account for diurnal variations.

### 2.5. Measurement Methods

#### 2.5.1. Hormone Level Measurement

To measure salivary concentrations of estradiol and progesterone, saliva was collected and analyzed using a saliva collection kit (SalivaBio A; Salimetrics, Carlsbad, CA, USA). Participants were asked to rinse the mouth prior to the start of the experiments, so that no food particles remained in the oral cavity. Saliva was collected at least 10 min after rinsing to prevent dilution of hormone concentrations in saliva. We also asked participants to strictly observe the following prohibitions and precautions during saliva collection: (1) no eating for at least 60 min prior to saliva collection; (2) no alcohol intake for at least 12 h prior to saliva collection; (3) no intake of sugary, acidic or caffeinated beverages prior to saliva collection; (4) no intake of dairy products for at least 20 min prior to saliva collection; (5) no tooth brushing for at least 45 min prior to saliva collection; and (6) no dental treatment within 48 h before saliva collection [26]. Saliva was collected in the mouth for 1 min, then transferred to a saliva collection container (Cryyovial; Salimetrics) using a special straw (Siva Collection Aid; Salimetrics). The saliva sample was immediately stored in a freezer at less than −80 °C. Analysis of female hormone concentrations was entrusted to Funakoshi Corporation (Tokyo, Japan). Samples were thawed at room temperature, mixed by vortexing, centrifuged at 1500× *g* for 15 min, and analyzed by enzyme-linked immunosorbent assay using high-sensitivity salivary immunoassay kits (17β-Estradiol Enzyme Immunoassay Kit and Salivary Progesterone Enzyme Immunoassay Kit; Salimetrics). The dilution factor was uniformly 1-fold (undiluted).

#### 2.5.2. Laxity Measurement

AKL was measured as the amount of anterior displacement of the tibia relative to the femur when loads of 67 N, 89 N, 111 N, and 133 N applied to the tibia. AKL was measured using a cruciate ligament function tester (KS Measure KSM-100; Japan Sigmax, Tokyo, Japan), and was performed only on the pivot (non-dominant) foot. Participants were placed in a supine position with the knee joint set in approximately 30° of flexion using a goniometer (Goniometer; Nishikawashinwa, Tokyo, Japan). The knee support was placed on the posterior part of the distal thigh and the foot support was placed under the foot. The position of the cruciate ligament function tester was adjusted so that the patellar contact point was at the center of the knee and the ankle fixation point was at the center of the ankle. The ankle was fixed with a lower limb fixation belt and the lower leg was fixed with a traction belt. After the participants was instructed to relax, measurement was performed by operating the load handle. Five measurements were taken and after discarding the maximum and minimum values, the average of the three remaining measurements was recorded. Measurement of knee joint angle and operation of the load handle were performed by a single researcher (S.M.). The intra-rater reliability of AKL measurement was confirmed to be higher than in our previous study [17]. Stiffness was calculated as change in (Δ)force/Δanterior displacement at 67–89 N, 89–111 N, and 111–133 N loads [17].

GJL was measured using the Beighton method [27]. Mobility was measured at five locations: fifth (little) finger, wrist, elbow, knee, and spine. Assessment criteria were as follows: (1) extension of the fifth finger greater than 90°; (2) ability to touch the thumb to the forearm, (3) hyperextension of the elbow greater than 10°; (4) hyperextension of the knee greater than 10°; and (5) ability to perform forward flexion of the trunk with palm touching the floor in full extension of the knee joint. Criteria (1) to (4) were evaluated as 1 point for each side, and Criterion 5 was evaluated as 1 point, for a total of 9 points. GJL was measured using the goniometer for items based on joint angle. GJL measurements were performed by one researcher (S.M.).

GR was measured using a hyperextension apparatus (Takei Scientific Instrument Co., Niigata, Japan) to evaluate the range of motion of knee extension (Figure 3). Participants were seated in a long sitting position with the hip joint set in about 70° of flexion with both upper limbs behind them (Figure 3A). Knee joint extension was defined as 0° when the scales for seat height and foot height were the same. The distance from the knee to the heel was adjusted by placing the heel on the footrest (Figure 3B), and the right and left positions of the lower limbs were adjusted to achieve 0° of hip adduction (Figure 3C). The proximal patella was fixed with a knee-fixation belt, and foot width was also fixed. Participants were instructed in advance to signal just before the appearance of pain or discomfort in the knee joint, and the knee joint was extended by turning the foot elevation screw at a rate of approximately 1°·s^−1^ or less until the signal was given. At the time the participant gave the signal, the foot was held for 10 s and the value on the scale for foot height was recorded. Maximum angle of passive knee joint extension was later calculated using the change in foot height (H) and the length from the center of the knee to the heel (L) according to the following equation:GR [°] = Arctan (H/L) × 57.2958 

After one practice test, five measurements were taken. After discarding the maximum and minimum values, the average of the three remaining measurements was recorded. A rest period of 10 s was provided between each of the five trials. Measurements were taken only for the pivot (non-dominant) foot. GR measurements were performed by one researcher (S.M.).

### 2.6. Reliability of GR Measurements

The reliability of GR measurements was examined in 10 knees of 5 adult males (mean age, 23.4 ± 0.8 years; height, 168.1 ± 5.7 cm; weight, 60.9 ± 4.0 kg) and 10 knees of 5 adult females (mean age, 21 ± 0.9 years; height, 166.7 ± 8.6 cm; weight, 59.3 ± 11.9 kg) in this study, none of whom had orthopedic diseases or pain in the lower limbs. GR was measured using the methods described above, with an interval of at least 10 min between measurements by different researchers (S.M. and C.S.) to determine inter-rater reliability and an interval of at least 30 min between measurements by the same researcher (S.M.) for intra-rater reliability, on the same day. Intra- and inter-rater reliabilities were calculated using intraclass correlation coefficient (ICC) (1, 3) and (2, 3), respectively.

### 2.7. Statistical Analysis

Statistical analyses were performed using SPSS version 27.0 (IBM Corp, Tokyo, Japan). Split-plot repeated-measures analysis of variance (ANOVA) was conducted to com-pare salivary estradiol and progesterone concentrations, AKL, stiffness, GJL, and GR menstrual cycle phases in female non-athletes and female athletes, and to compare eumenor-rhea and oligomenorrhea groups (subject factors [eumenorrhea group, oligomenorrhea group] vs. cycle phase factors [early follicular phase, late follicular phase, ovulation phase, luteal phase]). When an interaction or main effect of the subject factor or the cycle phase factor was identified, one-way repeated-measures ANOVA was used for comparison within cycle phase factors, and statistical processing was performed by the Bonferroni method as a post-test. In addition, statistical processing was performed by the independent samples *t*-test and Mann-Whitney U test as a post-test for comparison within subject factors. Probability values of 5% were considered statistically significance.

## 3. Results

### 3.1. Female Non-Athletes

In salivary estradiol, split-plot repeated-measures ANOVA showed no interactions and no main effects of subject factor, but a main effect of the cycle phase factor [F (2.197, 37.343) = 7.23, *p* = 0.002, ηp2 = 0.30]. The results of Bonferroni post hoc testing showed that salivary estradiol concentrations were significantly higher in the ovulation and luteal phases than in the early follicular phase (*p* = 0.006 and *p* = 0.002, respectively) (Table 1).

In salivary progesterone, split-plot repeated-measures ANOVA showed an interaction [F (1.345, 22.872) = 3.97, *p* = 0.048, ηp2 = 0.19]. One-way repeated-measures ANOVA and Bonferroni post hoc testing showed that salivary progesterone concentrations was significantly higher in the luteal phase than in the early follicular phase and ovulation phases in the eumenorrhea group (*p* = 0.026 and *p* = 0.040, respectively). Results from the independent sample *t*-test and post hoc Mann-Whitney U-test showed that salivary progesterone concentrations were significantly higher in the eumenorrhea group than in the oligomenorrhea group during the luteal phase (*p* = 0.035) (Table 1).

In AKL, the results of split-plot repeated-measures ANOVA showed no interactions or main effects of subject or cycle-phase factors at any loadings (Table 2).

In stiffness, split-plot repeated-measures ANOVA showed no interactions and no main effects of cycle-phase factors between any loadings, but a main effect of the subject factor was seen only between 89–111 N [F (1, 17) = 6.61, *p* = 0.020, ηp2 = 0.28] and 111–133 N [F (1, 17) = 9.48, *p* = 0.007, ηp2 = 0.36] loadings. The results of the Bonferroni post hoc testing showed that stiffness at 89–111 N and 111–133 N was significantly higher in the oligomenorrhea group than in the eumenorrhea group (*p* = 0.020 and *p* = 0.007, respectively) (Table 3).

The results of split-plot repeated-measures ANOVA about GJL and GR showed no interactions or main effects of subject or cycle-phase factors (Table 4 and Table 5).

### 3.2. Female Athletes

In salivary estradiol, split-plot repeated-measures ANOVA showed no interactions and no main effects of subject factor, but a main effect of the cycle-phase factor [F(3, 39) = 5.33, *p* = 0.004, ηp2 = 0.29]. The results of Bonferroni post hoc testing showed that salivary estradiol concentrations were significantly higher in the luteal phase than in the early follicular phase (*p* = 0.039) (Table 1).

In salivary progesterone, split-plot repeated-measures ANOVA showed no interactions and no main effects of subject factor, but a main effect of the cycle-phase factor [F(1.500, 19.502) = 14.38, *p* = 0.0004, ηp2 = 0.53]. The results of Bonferroni post hoc testing showed that salivary progesterone concentrations were significantly higher in the luteal phase than in the early follicular, late follicular, and ovulation phases (*p* = 0.005, *p* = 0.001 and *p* = 0.022, respectively) (Table 1).

The results of split-plot repeated-measures ANOVA about AKL, stiffness, and GJL showed no interactions or main effects of subject or cycle-phase factors at any loadings (Table 2, Table 3 and Table 4).

In GR, split-plot repeated-measures ANOVA showed no interactions and no main effects of the subject factor, but a main effect of the cycle-phase factor [F (1.929, 25.074) = 13.32, *p* = 0.0001, ηp2 = 0.51]. The results of Bonferroni post hoc testing showed that GR was significantly higher in the late follicular, ovulation, and luteal phases than in the early follicular phase (*p* = 0.050, *p* = 0.011 and *p* = 0.004, respectively) (Table 5).

### 3.3. Reliability of GR Measurement

ICC (1, 3) was 0.828 and ICC (2, 3) was 0.854. According to the criteria of Landis et al. [28], reliability is considered “almost perfect” for ICCs of 0.81 or more, so the reliabilities of GR measurements in this study were considered almost perfect.

## 4. Discussion

To the best of our knowledge, this represents the first study to examine changes in joint laxity during the menstrual cycle among females with menstrual irregularities. The main findings of this study were that stiffness was significantly higher in the oligomenorrhea group than in the eumenorrhea group, although no significant difference in stiffness was seen between cycle phases when limited to female non-athletes. GR was significantly higher in the late follicular, ovulation, and luteal phases than in the early follicular phase, although no difference between groups was seen when limited to female athletes. No significant differences in AKL or GJL were between groups or cycle phases in either female non-athletes or female athletes.

Stiffness did not change with the menstrual cycle in female non-athletes; however, it was significantly higher in the oligomenorrhea group than the eumenorrhea. Karageanes et al. considered that repetition of the menstrual cycle (hormonal stimulation) may exert long-term effects on connective tissues, resulting in changes to estradiol receptor sensitivity or an unraveling effect in which ligaments gradually become looser [16]. In this study, the duration of exposure to estradiol may have been shorter in the oligomenorrhea group with longer menstrual cycles than in the eumenorrhea group. If the duration of exposure to estradiol affected the tensile properties of the ACL, stiffness may have been higher in the oligomenorrhea group than in the eumenorrhea group. Stiffness can be calculated and quantified from the load-displacement curve obtained from AKL measurements and has been suggested to be related to clinical end-feel [29]. Three phases are observed in the load-displacement curve: early phase: middle phase: and late phase, the late phase is considered to represent the terminal stiffness of the joint in which the ACL is fully involved in restraining tibial anterior displacement [29]. Davey et al. reported that a 1-standard deviation (SD) decrease in ACL stiffness was associated with a 2.37-fold increased risk of contralateral injury after initial ACL injury [30]. In addition, the incidence of ACL injury is reportedly higher among women with normal menstruation than among those using OCs [31]. Such considerations suggest that female non-athletes with oligomenorrhea may have fewer risk factors for ACL injury than female non-athletes with eumenorrhea.

GR in female athletes was significantly higher in the late follicular, ovulation, and luteal phases than in the early follicular phase, although no difference was evident between eumenorrhea and oligomenorrhea groups. These results support previous findings that GR changes during the menstrual cycle [12,18]. Estradiol receptors are present in the human ACL [32], myofibers of skeletal muscle and capillary endothelial cells [33]. Previous studies investigating changes in muscle stiffness during the menstrual cycle have reported that hamstring extensibility (straight leg raising angle) was higher during the ovulation phase compared to the early follicular phase [34], and muscle stiffness of the vastus medialis and semitendinosus muscles was higher during the ovulation phase compared to the luteal phase [35]. Avrillon et al. reported that the shear modulus of the semimembranosus muscle was lower among figure skaters, taekwondo practitioners, soccer players, and fencers than among non-athletes [36]. This suggests that female athletes with lower muscle stiffness may be more susceptible to changes in joint laxity than non-athletes. Therefore, it is possible that estradiol may have altered muscle stiffness, resulting in altered GR in female athletes during the menstrual cycle. The results of this study thus suggest that female athletes may be at increased risk of ACL injury from the late follicular phase to the luteal phase. In the future, a better understanding of the menstrual cycle of female athletes will be needed to develop training methods adapted to the cyclical changes in body structure and methods for preventing ACL injury.

The AKL did not change during the menstrual cycle in either female non-athletes or female athletes, and no difference was seen between eumenorrhea and oligomenorrhea groups. The results of this study support previous studies that AKL does not change during the menstrual cycle [8,15,16,17]. Lee et al. found that AKL was significantly higher in the non-OC group than in the OC group and that AKL changed only in the non-OC group [19]. In the present study, no difference in estradiol concentration was identified between eumenorrhea and oligomenorrhea groups, suggesting that no group differences may exist in AKL. GJL did not change during the menstrual cycle in either female non-athletes or female athletes, and no difference was apparent between eumenorrhea and oligomenorrhea groups. The results of this study differed from those of previous studies, which showed that GJL changed during the menstrual cycle [12,18]. GJL can be congenital, occurring as a part of connective tissue diseases such as Marfan’s syndrome, Ehlers-Danlos syndrome, and benign joint hypermobility syndrome [37], or acquired, resulting from stretching of the capsular ligament due to repetitive microtrauma or repetitive use during sports activities [38]. GJL is thus suggested to be highly susceptible to congenital laxity or sports history, but less susceptible to female hormones during the menstrual cycle.

Several limitations to the present study must be considered when interpreting the results. First, considering the characteristics of the experimental apparatus and the safety of participants, GR was evaluated based on the subjective sensations of participants. In this study, GR was measured by extending the knee joint until just before the appearance of pain or discomfort in the knee joint, so pain or intrinsic receptive sensation may have been affected. The pain threshold reported does not change during the menstrual cycle [39], but intrinsic receptive sensation in the knee joint is reported to decrease in the early follicular phase compared to the luteal phase [40]. However, the reliability of GR measurement appeared to be high in this study, so the method of measuring GR in this study was considered to be highly valid. A second limitation was that we could not control the amount of activity of female athletes. In this study, female athletes were tested throughout a single season, because the study period was extended due to the coronavirus disease 2019 pandemic. The activity levels of female athlete participants may thus have differed between training and competition periods, which may have contributed to discrepancies in the results for AKL, GJL, and GR. Therefore, when assessing joint laxity in female athletes, consideration must be given to not only the effects of the menstrual cycle, but also the amount of activity during the season. A third limitation was the sample size. In the present study, joint laxity was measured eight times in four phases: early follicular phase, late follicular phase, ovulation phase, and luteal phase. Of the 71 female non-athletes and 27 female athletes recruited, only 19 female non-athletes and 15 female athletes completed the experiment. Some degree of caution is thus warranted when interpreting the present results. However, by measuring joint laxity at multiple time points throughout the menstrual cycle, we were able to more accurately investigate changes in joint laxity during the menstrual cycle. A fourth limitation was that the participants were limited to university students and university athletes. Previous studies have reported that age may affect the value of joint laxity [10,41]. Therefore, the results of this study may not be applicable to non-university age groups. Future studies should be conducted not only in university age group but also in middle and high school age groups.

## 5. Conclusions

In this study, stiffness was significantly higher in the oligomenorrhea group than in the eumenorrhea group when limited to female non-athletes. GR was significantly higher in the late follicular phase, ovulation phase, and luteal phase compared to the early follicular phase when limited to female athletes. Future studies need to clarify the effects of the menstrual cycle in female athletes and develop training methods for injury prevention that are adapted to cyclical changes in female body structure.

## Figures and Tables

**Figure 1 jcm-11-03222-f001:**
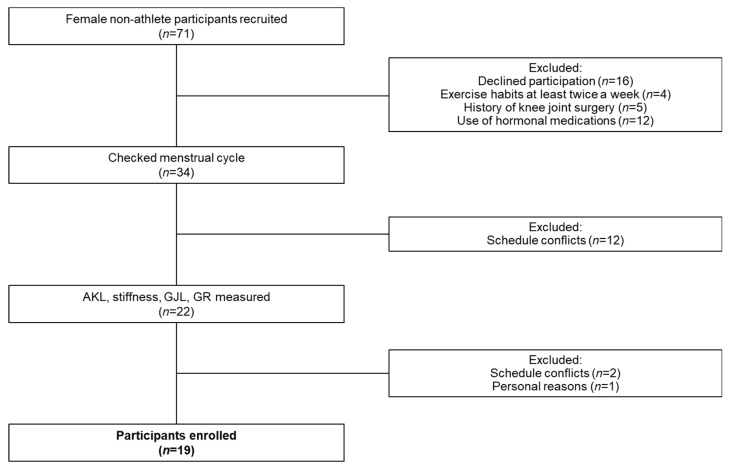
Flowchart for selection of female non-athlete participants. AKL, anterior knee laxity; GJL, general joint laxity; GR, genu recurvatum.

**Figure 2 jcm-11-03222-f002:**
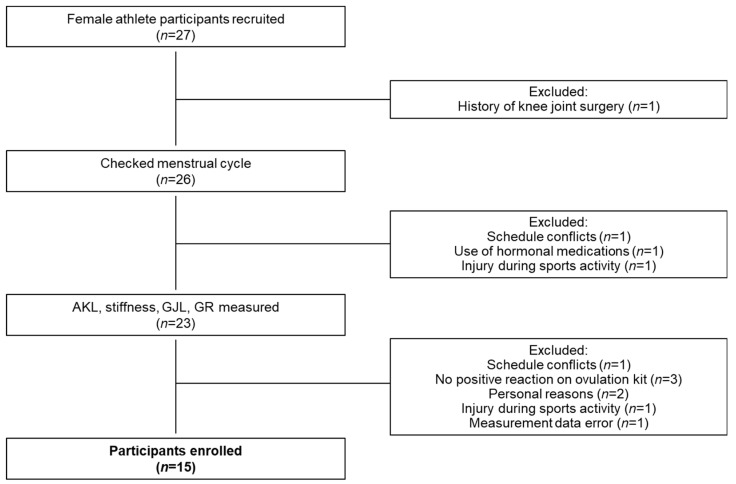
Flowchart for selection of female athlete participants. AKL, anterior knee laxity; GJL, general joint laxity; GR, genu recurvatum.

**Figure 3 jcm-11-03222-f003:**
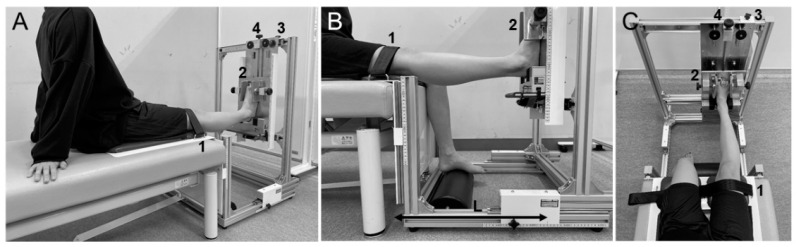
Position for measurement of genu recurvatum. 1: Knee-fixation belt; 2: foot width fixing part; 3: right/left adjustment of lower limbs; 4: foot elevation screw; L, length from center of knee to heel. The participant was seated on the hyperextension apparatus (Takei Scientific Instrument Co., Niigata, Japan) in a long sitting position with the hip joint set in about 70° of flexion with both upper limbs behind them (**A**). Distance from the knee to the heel was adjusted by placing the heel on the footrest (**B**), and positions of the lower limbs were adjusted to the left/right to achieve 0° of hip adduction (**C**). The proximal patella was fixed with the knee-fixation belt, and foot width was also fixed. Participants were instructed in advance to signal just before the appearance of pain or discomfort in the knee joint, and the knee joint was extended by turning the foot elevation screw at a rate of approximately 1°·s^−1^ or less until that signal was given. At the time the signal was given, foot height was recorded. Maximum angle of passive knee joint extension was calculated later based on the result recorded.

**Table 1 jcm-11-03222-t001:** Changes in estradiol and progesterone concentrations during the menstrual cycle.

	Early Follicular Phase	Late Follicular Phase	Ovulation Phase	Luteal Phase	Total
Estradiol [pg/mL]					
Female non-athletes (*n* = 19)	1.1 ± 0.3	1.3 ± 0.4	1.5 ± 0.4 *^a^*	1.4 ± 0.4 *^b^*	
Eumenorrhea group (*n* = 11)	1.1 ± 0.3	1.3 ± 0.4	1.5 ± 0.4	1.5 ± 0.4	1.3 ± 0.4
Oligomenorrhea group (*n* = 8)	1.0 ± 0.3	1.2 ± 0.4	1.5 ± 0.4	1.4 ± 0.3	1.3 ± 0.4
Female athletes (*n* = 15)	1.0 ± 0.3	1.0 ± 0.3	1.2 ± 0.3	1.3 ± 0.6 *^f^*	
Eumenorrhea group (*n* = 8)	0.8 ± 0.2	0.9 ± 0.3	1.2 ± 0.3	1.1 ± 0.6	1.0 ± 0.4
Oligomenorrhea group (*n* = 7)	1.2 ± 0.2	1.2 ± 0.3	1.1 ± 0.3	1.5 ± 0.4	1.3 ± 0.3
Progesterone [pg/mL]					
Female non-athletes (*n* = 19)	132.0 ± 75.3	161.1 ± 77.4	164.5 ± 94.5	343.7 ± 280.2	
Eumenorrhea group (*n* = 11)	146.7 ± 81.3	167.1 ± 85.4	172.1 ± 104.0	477.6 ± 324.4 *^c,d,e^*	233.4 ± 148.8
Oligomenorrhea group (*n* = 8)	111.7 ± 65.9	152.9 ± 69.7	154.1 ± 85.3	200.7 ± 105.8	154.8 ± 81.7
Female athletes (*n* = 15)	127.6 ± 61.0	141.2 ± 60.3	147.2 ± 73.1	231.4 ± 108.6 *^g,h,i^*	
Eumenorrhea group (*n* = 8)	101.1 ± 66.2	146.7 ± 70.3	172.1 ± 83.1	242.1 ± 128.5	165.5 ± 87.0
Oligomenorrhea group (*n* = 7)	157.8 ± 39.8	135.0 ± 51.3	118.8 ± 51.5	219.2 ± 89.0	157.7 ± 57.9

Values are presented as means ± SD. *^a^*—Statistically significant difference compared with early follicular phase in female non-athletes (*p* = 0.006). *^b^*—Statistically significant difference compared with early follicular phase in female non-athletes (*p* = 0.002). *^c^*—Statistically significant difference compared with early follicular phase in the eumenorrhea group of female non-athletes (*p* = 0.026). *^d^*—Statistically significant difference compared with ovulation phase in the eumenorrhea group of female non-athletes (*p* = 0.040). *^e^*—Statistically significant difference compared with oligomenorrhea in the luteal phase of female non-athletes (*p* = 0.035). *^f^*—Statistically significant difference compared with the early follicular phase in female athletes (*p* = 0.039). *^g^*—Statistically significant difference compared with the early follicular phase in female athletes (*p* = 0.005). *^h^*—Statistically significant difference compared with the late follicular phase in female athletes (*p* = 0.001). *^i^*—Statistically significant difference compared with the ovulation phase in female athletes (*p* = 0.022).

**Table 2 jcm-11-03222-t002:** Changes in anterior knee laxity during the menstrual cycle.

		Early Follicular Phase	Late Follicular Phase	Ovulation Phase	Luteal Phase	Total
Anterior knee laxity [mm]						
Female non-athletes (*n* = 19)						
67 N	Eumenorrhea group (*n* = 11)	4.6 ± 1.3	4.6 ± 1.4	4.4 ± 1.2	4.2 ± 1.2	4.5 ± 1.3
	Oligomenorrhea group (*n* = 8)	4.5 ± 2.1	4.4 ± 1.7	4.5 ± 1.4	4.3 ± 1.5	4.4 ± 1.7
89 N	Eumenorrhea group (*n* = 11)	5.7 ± 1.5	5.6 ± 1.6	5.3 ± 1.3	5.2 ± 1.2	5.5 ± 1.4
	Oligomenorrhea group (*n* = 8)	5.4 ± 2.2	5.3 ± 1.8	5.4 ± 1.5	5.1 ± 1.4	5.3 ± 1.7
111 N	Eumenorrhea group (*n* = 11)	6.6 ± 1.6	6.5 ± 1.8	6.2 ± 1.4	6.0 ± 1.2	6.3 ± 1.5
	Oligomenorrhea group (*n* = 8)	6.0 ± 2.3	6.0 ± 1.8	6.1 ± 1.5	5.8 ± 1.5	6.0 ± 1.8
133 N	Eumenorrhea group (*n* = 11)	7.5 ± 1.7	7.4 ± 1.9	7.0 ± 1.5	6.8 ± 1.3	7.2 ± 1.6
	Oligomenorrhea group (*n* = 8)	6.7 ± 2.3	6.7 ± 1.8	6.8 ± 1.5	6.4 ± 1.5	6.6 ± 1.8
Female athletes (*n* = 15)						
67 N	Eumenorrhea group (*n* = 8)	3.6 ± 1.2	3.8 ± 1.0	3.8 ± 1.0	3.4 ± 0.9	3.6 ± 1.0
	Oligomenorrhea group (*n* = 7)	4.0 ± 2.0	4.5 ± 1.3	4.0 ± 1.7	4.0 ± 1.5	4.1 ± 1.6
89 N	Eumenorrhea group (*n* = 8)	4.5 ± 1.4	4.7 ± 1.3	4.7 ± 1.1	4.1 ± 1.1	4.5 ± 1.2
	Oligomenorrhea group (*n* = 7)	4.8 ± 2.2	5.4 ± 1.6	4.8 ± 1.9	4.9 ± 1.7	5.0 ± 1.9
111 N	Eumenorrhea group (*n* = 8)	5.2 ± 1.5	5.4 ± 1.5	5.4 ± 1.2	4.8 ± 1.2	5.2 ± 1.4
	Oligomenorrhea group (*n* = 7)	5.5 ± 2.2	6.1 ± 1.8	5.5 ± 2.1	5.7 ± 2.0	5.7 ± 2.0
133 N	Eumenorrhea group (*n* = 8)	5.9 ± 1.6	6.0 ± 1.7	6.1 ± 1.4	5.4 ± 1.2	5.9 ± 1.5
	Oligomenorrhea group (*n* = 7)	6.2 ± 2.4	6.8 ± 2.0	6.1 ± 2.1	6.4 ± 2.2	6.4 ± 2.2

Values are presented as means ± SD.

**Table 3 jcm-11-03222-t003:** Changes in stiffness during the menstrual cycle.

		Early Follicular Phase	Late Follicular Phase	Ovulation Phase	Luteal Phase	Total
Stiffness [N/mm]						
Female non-athletes (*n* = 19)						
67–89 N	Eumenorrhea group (*n* = 11)	21.6 ± 4.3	23.4 ± 5.6	24.5 ± 6.3	24.6 ± 2.9	23.5 ± 4.8
	Oligomenorrhea group (*n* = 8)	27.5 ± 7.0	27.1 ± 5.7	25.2 ± 5.8	28.4 ± 6.0	27.1 ± 6.1
89–111 N	Eumenorrhea group (*n* = 11)	25.6 ± 5.5	26.5 ± 6.6	28.4 ± 7.7	27.6 ± 3.5	27.0 ± 5.8
	Oligomenorrhea group (*n* = 8)	34.3 ± 9.6	33.6 ± 9.0	30.6 ± 5.4	35.6 ± 3.9	33.6 ± 7.0 *^a^*
111–133 N	Eumenorrhea group (*n* = 11)	26.7 ± 6.9	27.0 ± 5.1	26.9 ± 4.5	28.1 ± 4.3	27.2 ± 5.2
	Oligomenorrhea group (*n* = 8)	35.5 ± 8.9	35.9 ± 10.7	32.3 ± 6.6	38.0 ± 7.4	35.4 ± 8.4 *^b^*
Female athletes (*n* = 15)						
67–89 N	Eumenorrhea group (*n* = 8)	26.8 ± 8.1	28.8 ± 7.8	27.9 ± 5.1	28.3 ± 12.4	28.0 ± 0.4
	Oligomenorrhea group (*n* = 7)	31.5 ± 10.3	29.1 ± 12.8	31.9 ± 14.3	27.3 ± 11.4	29.9 ± 12.2
89–111 N	Eumenorrhea group (*n* = 8)	32.8 ± 9.0	34.2 ± 9.2	32.5 ± 7.7	36.2 ± 10.6	33.9 ± 9.1
	Oligomenorrhea group (*n* = 7)	33.2 ± 7.8	36.3 ± 15.7	36.7 ± 11.4	32.8 ± 14.3	34.8 ± 12.3
111–133 N	Eumenorrhea group (*n* = 8)	37.1 ± 11.5	38.1 ± 11.9	36.7 ± 11.5	38.0 ± 7.1	37.5 ± 10.5
	Oligomenorrhea group (*n* = 7)	34.6 ± 9.3	40.0 ± 21.9	43.6 ± 18.9	32.3 ± 13.1	37.7 ± 15.8

Values are presented as means ± SD. *^a^*—Statistically significant difference compared with the eumenorrhea group (*p* = 0.020). *^b^*—Statistically significant difference compared with the eumenorrhea group (*p* = 0.007).

**Table 4 jcm-11-03222-t004:** Changes in general joint laxity during the menstrual cycle.

	Early Follicular Phase	Late Follicular Phase	Ovulation Phase	Luteal Phase	Total
General joint laxity [points]					
Female non-athletes (*n* = 19)					
Eumenorrhea group (*n* = 11)	1.7 ± 1.4	1.6 ± 1.4	1.4 ± 1.2	1.4 ± 1.3	1.5 ± 1.3
Oligomenorrhea group (*n* = 8)	1.6 ± 1.3	1.6 ± 1.1	1.3 ± 1.3	1.5 ± 1.0	1.5 ± 1.2
Female athletes (*n* = 15)					
Eumenorrhea group (*n* = 8)	2.2 ± 1.7	2.3 ± 1.5	2.5 ± 1.4	2.6 ± 1.6	2.4 ± 1.5
Oligomenorrhea group (*n* = 7)	1.6 ± 2.5	1.9 ± 2.6	1.7 ± 2.9	1.6 ± 2.6	1.7 ± 2.6

Values are presented as means ± SD.

**Table 5 jcm-11-03222-t005:** Changes in genu recurvatum during the menstrual cycle.

	Early Follicular Phase	Late Follicular Phase	Ovulation Phase	Luteal Phase	Total
Genu recurvatum [°]					
Female non-athletes (*n* = 19)	7.0 ± 3.9	7.2 ± 3.9	7.3 ± 4.1	7.7 ± 4.1	
Eumenorrhea group (*n* = 11)	6.2 ± 3.9	6.3 ± 3.8	6.3 ± 3.9	6.6 ± 4.1	6.4 ± 3.9
Oligomenorrhea group (*n* = 8)	8.1 ± 3.9	8.4 ± 3.8	8.6 ± 4.2	9.3 ± 3.9	8.6 ± 3.9
Female athletes (*n* = 15)	7.7 ± 3.7	8.3 ± 3.7 *^a^*	8.7 ± 3.8 *^b^*	9.0 ± 3.8 *^c^*	
Eumenorrhea group (*n* = 8)	6.1 ± 3.7	7.0 ± 4.1	7.4 ± 4.4	7.6 ± 4.3	7.0 ± 4.1
Oligomenorrhea group (*n* = 7)	9.5 ± 2.8	9.8 ± 2.8	10.2 ± 2.6	10.7 ± 2.4	10.1 ± 2.6

Values are presented as means ± SD. *^a^*—Statistically significant difference compared with the early follicular phase in female athletes (*p* = 0.050). *^b^*—Statistically significant difference compared with the early follicular phase in female athletes (*p* = 0.011). *^c^*—Statistically significant difference compared with the early follicular phase in female athletes (*p* = 0.004).

## Data Availability

The data that support the findings of this study are available from the corresponding author.

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
