# Peer review of "Menstrual Cycle Changes Joint Laxity in Females—Differences between Eumenorrhea and Oligomenorrhea"

_jcm, 2022, doi:10.3390/jcm11113222_

Round 1
Reviewer 1 Report
The manuscript is well written. I have few minor comments
‘However, the rate of OC use among female athletes is known to be lower in Japan than 72 that in Western countries [25-28]’ - Why this statement is required in the introduction.is this study a regional-specific study?
How the sample size was calculated?
The exact name of the ethical committee is required. Don’t write our institute
The result section has to be improved. Why same sentences are being repeated.
The discussion is written well
Reviewer 2 Report
This study investigates the effect of the menstrual cycle on joint laxity in females differences between eumenorrhea and oligomenorrhea. The results of this study provide the rationale for setting up different injury prevention protocols according to the menstrual cycle in female athletes. However, as similar contents are repeated in the introduction and discussion, the text is lengthy and it is not easy for readers to understand. If the composition is changed a little more briefly and readability is improved, it is judged that it will be a good research paper.
1.Line 47-65 In the Introduction, too detailed explanation is rather confusing. It is better to summarize it in 2 or 3 sentences and discuss the details in the discussion.
2. Line 72-73. It seems to be an unnecessary sentence of the flow of context. It would be better to delete it.
3. Line 74-75 The definition of “oligomenorrhea” was mentioned more details at line 116-118. I don't think it's necessary to mention it here.
4. Line 79-81 This sentence seems to be an excellent word to clarify novelty.
5. Line 96-98 Non-athletic 19 people / athletic 15 people (volleyball 10 people, basketball 5 people)
The muscles used for each exercise (especially the difference in Q strength) are different, and it is likely that stiffness may be affected depending on the exercise pattern. Shouldn't this be taken into account?
6. Line 157/Line 176 Please add sub-title (e.g 2.5.1 Hormone level measurement / 2.5.2 Laxity measurement)
7.Line 268-271. The Bonferroni method is to adjust after one-way repeated-measures ANOVA, the significance probability of the post hoc test by the number of iterations. Still, it seems appropriate to change the significant value further down compared to 5%.
8.Line 262-271. In my opinion, the number of cases in each group and sub-group is too small to test statistical significance. Could you provide data of statistical power?
9. Line 283-286. After applying one-way repeated-measures ANOVA and Bonferroni post hoc adjustment, the significant level should be adjust considering repeated times of M-W U test.
(Ex. 3 times of M-W U-test, the significant level 0.05/3) Please check authors statistical methods.
10. Line 382-384. There is no need for repeated comments about the topics at the discussion. please delete first sentence in the “discussion section”
11. Line 393-394 How about modifying to “Stiffness did not change with the menstrual cycle in female non-athletes; however, it was significantly higher in the oligomenorrhea group than in the eumenorrhea group.”?
12. Line 421-424 Since this study focuses on anterior laxity, it is better not to mention posterior laxity.
13. Line 435. The sentence already commented previously.
14. Line 418~ Why do the authors think that GR increases during the late follicular phase and ovulation phase only in the athlete group? It is necessary to consider this part.
15. Overall, the discussion is too verbose and complicated, which further confuses the content of the message. It would be better to shorten it to 4 paragraphs and re-organize it.
Round 2
Reviewer 2 Report
The concerns related to the recommended modifications have been well corrected.
It is expected to add depth of knowledge to the development of injury prevention programs for athletes related to the menstrual cycle and ligament laxity.